# *Penicillium* spp. XK10, Fungi with Potential to Repair Cadmium and Antimony Pollution

Yiying He [1], Chaoyang Li [1], Zhongyu Sun [2], Wan Zhang [1], Jianing He [2], Yunlin Zhao [1,*], Zhenggang Xu [1,2,*] and Weiping Zhao [3]

[1] Hunan Research Center of Engineering Technology for Utilization of Environmental and Resources Plant, Central South University of Forestry and Technology, Changsha 410004, China

[2] Key Laboratory of National Forestry and Grassland Administration on Management of Western Forest Bio-Disaster, College of Forestry, Northwest A&F University, Yangling 712100, China

[3] School of Business, Hunan Agricultural University, Changsha 410128, China

* Correspondence: zyl8291290@163.com (Y.Z.); xuzhenggang@nwafu.edu.cn (Z.X.)

**Abstract:** Soil heavy-metal pollution is one of the most important environmental problems in the world, and seriously endangers plant growth and human health. Microbial remediation has become a key technology in the field of soil heavy-metal remediation due to its advantages of being harmless, green and environmental. In this study, a fungus *Penicillium* spp. XK10 with high tolerance to cadmium (Cd) and antimony (Sb) was screened from mine slag, and its adsorption characteristics to heavy metals under different environmental conditions were studied. The results showed that at $pH_0 = 6$, $C_0$ (Cd) = 0.1 mM, and the adsorption time was 4 days, the maximum removal rate of cadmium by XK10 was 32.2%. Under the conditions of $pH_0 = 4$, T = 7d, and the initial antimony concentration of 1 mM, the removal rate of antimony by XK10 was the highest, which was 15.5%. This study provides potential microbial materials for bioremediation of heavy metal-contaminated soils.

**Keywords:** *Penicillium*; heavy metal pollution; Cd; Sb; biosorption





## 1. Introduction

With the intensification of industrialization, soil heavy-metal pollution has become a global environmental issue [1,2]. At present, the research into soil heavy-metal pollution mainly focuses on mercury (Hg), lead (Pb), cadmium (Cd), chromium (Cr), zinc (Zn) and other heavy metals, while the research on antimony (Sb) is relatively less [3]. The residual heavy metals in the soil not only reduce soil fertility, but also pose a great threat to human health through the food chain [4]. It is a very urgent task to develop heavy metal pollution-control technology [5]. Compared with physical remediation and chemical remediation, bioremediation has the advantages of low cost and environmental friendliness, and has become a promising heavy metal remediation method [6]. Microorganisms play an important role in bioremediation. They can not only control heavy metal pollution alone, but also enhance the remediation effect of plants on heavy metal pollution [7–9]. Currently, most of the research is aimed at the single adsorption of heavy metals by microorganisms, while there are few studies on the simultaneous adsorption of multiple heavy metals by microorganisms [10]. In the actual heavy metal pollution cases, the phenomenon of multiple heavy metals causing complex pollution is more common, so we should pay attention to the adsorption capacity of microorganisms for multiple heavy metals.

Fungi have the advantages of easy formation of mycelial spheres, large contact area, high biological adsorption capacity, rapid growth and good solid–liquid separation, which make them an ideal adsorption material [11–13]. In the field of fungal bioremediation, mycorrhizal fungi and yeasts are the hot spots of interest. Among the mycorrhizal fungi, *Penicillium* sp., *Aspergillus* sp., *Mucor* sp. and *Rhizopus* sp. have been studied more, and among the yeasts, *Saccharomyces* spp. have been studied more [14,15]. Numerous studies

have shown that *Penicillium* spp. was a potential adsorbent material. Xu et al. found that *P. chrysogenum* was a material for remediation of Cd-contaminated soil [16]. Tian et al. found that *P. oxalicum* and *A. niger* were two potential microbes to remediate Pb contamination [17]. Deng et al. found that *P. chrysogenum* strain F1 could remove heavy metals in contaminated soil [18]. Fourest et al. found that *P. chrysogenum* could grow in stressed environments and exhibits high resistance to and accumulation of heavy metals [19]. Heavy metal sorption by fungi is also influenced by various environmental factors (e.g., pH, heavy metal concentration and biomass) [20–23]. Therefore, a large number of studies have been conducted to screen for resistant fungi and to study their sorption characteristics on heavy metals under different environmental conditions.

The antimony mine in Lengshuijiang City (Hunan, China) is world-renowned for its abundant Sb reserves, and is known as the "World Sb Capital" [24]. Industrial mining has led to excessive Sb and Cd content in local soil and groundwater. Sb is not only carcinogenic to humans, but also has a long latency period, which can cause diseases such as lung inflammation and chronic bronchitis [25,26]. At the same time, Cd is one of the common toxic heavy metals, which is easy to migrate and be absorbed by plants. Cd can be transferred from the soil to plants and eventually accumulate in humans through the food chain [27]. In addition, high concentrations of Cd may cause many diseases, such as osteoporosis, cardiovascular, cerebrovascular diseases and other respiratory problems [28,29]. In general, due to the poor mobility, the long residence time and the inability to be degraded by microorganisms in soil, the heavy metal pollutants can eventually affect human health through media such as water and plants. In the mining area, the microbial resources are abundant, and some microorganisms have been resistant to heavy metals for a long time in the polluted environment. Therefore, the isolation and screening of tolerant strains in the mining area is more conducive to the treatment of heavy metal pollution and the prevention of ecological risks. However, the survival and repair effect of microorganisms will be restricted by environmental conditions such as temperature, pH and heavy metal concentration. In order to provide fungal materials for the treatment of Cd and Sb combined pollution widely existing in the environment, this study screened a Cd- and Sb-tolerant fungus from the antimony slag, and explored the adsorption characteristics of the fungus to the heavy metals under different environmental conditions. It not only enriched the biological materials for heavy metal pollution treatment, but also explored the influence of different heavy metals on the adsorption capacity of adsorption fungi.

## 2. Materials and Methods

### 2.1. Soil Samples Collected and Culture Medium Preparation

The antimony mine of Lengshuijiang City (Hunan, China) has been mined since 1892, amounting to more than 100 years. The reserves of antimony ore resources are more than 300,000 tons, accounting for 30% of the global proportion. Soil samples were collected in the antimony mine (27°45′16″ N; 111°29′21″ E) on 4 October 2018. The soil contains a variety of heavy metals, and most heavy metal content greatly exceeds the environmental background values, which may pose a risk to human health. Among them, the contents of Sb and Cd in the soil reach $11,753.0 \pm 176.1$ mg kg$^{-1}$ and $64.3 \pm 5.5$ mg kg$^{-1}$, respectively. In addition, the pH of the tailings slag is around 6.24, indicating that the tailings slag is a weakly acidic soil. Samples were collected from the slag waste site, and repeated three times. A total of 0–10 cm of tailings slag was selected, and after natural air-drying, the tailing slag was filtered through a 100 mesh sieve, divided into 50 mL centrifuge tubes, and then stored in a refrigerator at 4 °C for later use.

During the experiment, three culture media were used and their configuration methods were as follows. (1) Martin's solid medium: KH$_2$PO$_4$ (1 g), MgSO$_4$·7H$_2$O (5 g), peptone (5 g), glucose (10 g), water (1000 mL); (2) Potato dextrose broth (PDB) medium: peeled potato (200 g), sucrose (50 g), distilled water (1000 mL); (3) Potato dextrose agar (PDA) medium: peeled potatoes (200 g), sucrose (50 g), agar (20 g), distilled water (1000 mL). The three culture solutions were sterilized at $1 \times 10^5$ Pa for 25 min.

### 2.2. Isolation and Identification of Cd and Sb Tolerant Fungi

The isolation process of the tolerant fungi was as follows. Firstly, 10 g of naturally air-dried slag was weighed and added to 90 mL sterile water containing glass beads, and the culture was shaken at 120 r/min for 30 min. Then, the supernatant was diluted 10 times to make a soil suspension with a dilution of $10^{-2}$ g/mL. Pipetted 500 µL of the suspension into Martin's solid medium containing 0.5 mM Cd(II), inverted and incubated at 30 °C for 5–7 d. The strains with different colony morphology and well growth were selected, purified and numbered by the plate streaking method. The purified strains were transferred to Martin's solid medium containing 5 mM Cd(II) [30,31] again, and cultured under the same conditions for 7 d. The strains with better growth were regarded as Cd-tolerant fungi. Similarly, the Sb-tolerant fungi was inoculated into Martin's solid medium containing 7 mM Sb(III) for 7 d; the strains with better growth were cadmium- and antimony-tolerant fungi.

One of the screened strains was named XK10 and XK10 was picked using an inoculation loop and incubated in a conical flask containing 100 mL PDB medium, shaken at 120 rpm and 28 °C for 36 h. Then, 1 mL of suspension was added to fresh 100 mL PDB medium and was cultured under the same conditions. The biomass of the fungi was measured every 12 h until it remained constant. The growth curve of XK 10 was plotted according to the time and biomass, and the time when the fungi was in the logarithmic growth period was recorded.

Molecular identification was performed by Shanghai Majorbio Biomedical Technology Co., Ltd. in Shanghai, China. (http://www.majorbio.com, accessed on 6 December 2022) using the XK10 supernatant as the template. Firstly, the genomic DNA of the tolerant fungi was extracted, forward primer ITS1 (5′–TCCGTAGGTGAACCTGCGG–3′) and the reverse primer ITS4 (5′–TCCTCCGTTATTGATATGC–3′) were used to amplify the ITS region of the tolerant fungi by PCR. PCR reaction conditions: 98 °C for 2 min, 98 °C for 10 s, 54 °C for 10 s, 72 °C for 10 s, 35 cycles; and 72 °C for 5 min. Then, the PCR amplification products were subjected to sequence homology research through BLAST in the NCBI database, and the MrBayes program was used to construct the phylogenetic tree of the resistant fungi. At the same time, the potential of phosphate solubilization, IAA production and siderophore production were determined as in the reported method [9]. The ITS sequences were submitted to NCBI (National Center for Biotechnology Information) with accession number OP692700.

### 2.3. Determination of the Growth Curve and Tolerance

In order to understand the tolerance of XK 10 to Cd and Sb, respectively, the fungi in the logarithmic growth phase was selected. A total of 1 mL of spore suspension was added to 49 mL PDB medium in the Erlenmeyer flask (pH = 6) and incubated at 28 °C and 120 rpm for 0–7 days, respectively. The biomass of the fungi was measured every 24 h by dry weight method and stopped when the biomass was no longer increasing. The concentration of Cd solution was set to 0 Mm (control), 0.1 mM, 0.5 mM and 2 mM, respectively. The concentration of Sb solution was set to 0 Mm (control), 0.1 mM, 0.5 mM and 2 mM, respectively. At the same time, the potential of phosphate solubilization, IAA production and siderophore production were determined as in the reported method [9].

### 2.4. One-Factor Experiments Design

In order to understand the tolerance of different factors on the XK10 adsorption effect, the single factor experimental design was carried out. The adsorption capacity ($q_e$) and removal rate (Q) of XK10 were carried out according to the following statement. A total of 1 mL biosorbent was added to 49 mL PDB medium and incubated at 28 °C and 120 rpm until logarithmic phase. Then, the pH value ($Ph_{LP}$) and fungal biomass ($M_{LP}$) of the medium was determined; added 50 mL heavy metal solution (Cd or Sb) and measured the pH ($pH_0$). At the end of the experiment, the culture was harvested by filtration using Whatman No. 11 filter paper, and the dry weight was gathered. Two new fractions were obtained: the liquid media (Fraction A) and the dried mycelium (Fraction B). The pH and

metal concentration of Fraction A was measured. Biomass production (dried mass) was measured of Fraction B (Figure 1).

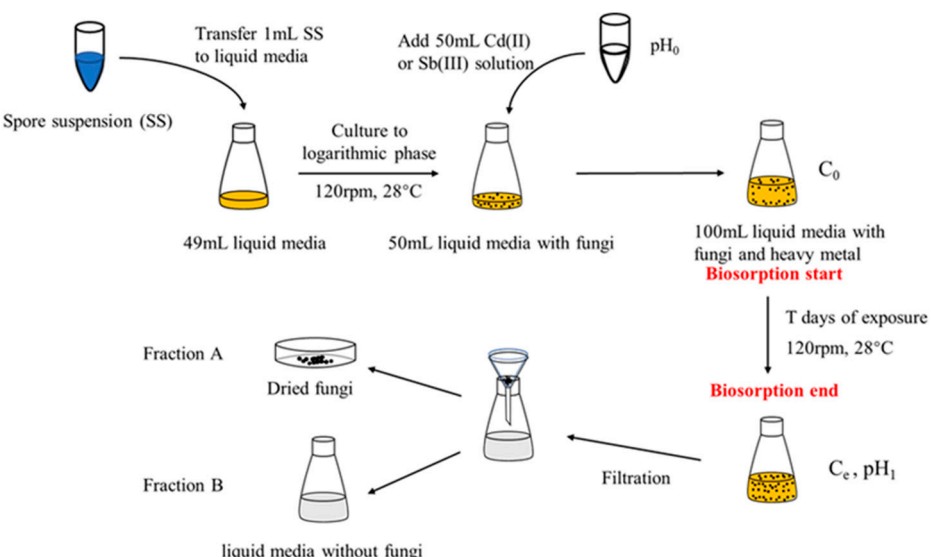

**Figure 1.** Schematic diagram of adsorption experiments of XK10.

The adsorption capacity ($q_e$) and removal rate (Q) of heavy metals Cd and Sb by XK10 were calculated according to Equations (1) and (2):

$$q_e = \frac{(C_0 - C_e)V}{M} \tag{1}$$

$$Q = \frac{C_0 - C_e}{C_0} \times 100\% \tag{2}$$

where $q_e$ is the weight of the heavy metal adsorbed by XK10 per unit weight ($mg \cdot g^{-1}$); $C_0$ and $C_e$ are the initial concentration of heavy metal ions in the solution and the concentration at adsorption equilibrium ($mg \cdot L^{-1}$), respectively. V is the volume of the heavy metal solution (mL). M is the biomass of fungi (g). Q is the removal rate of heavy metals by XK10 (%).

The adsorption experiments of heavy metals by XK10 were carried out under different environmental conditions. The initial concentrations of the heavy metal Cd solution ($C_0$ (Cd)) were set as 0.1, 0.3, 0.5 mM, and the initial concentrations of Sb solution ($C_0$ (Sb)) were 0, 1, 4, 7 mM. Initial pH ($pH_0$) of the solution was 2, 4, 6 [32,33], and the adsorption times (T) were 1, 4, 7 d. The effect of $C_0$ (Cd) on Cd adsorption by fungi was investigated under the conditions of $pH_0$= 6 and T= 7 d. The effect of adsorption time (T) on Cd adsorption by fungi was studied under the conditions of $pH_0$ = 6 and $C_0$ (Cd) = 0.1 mM. The effect of the initial pH on Cd adsorption by fungi was studied under the conditions of T = 7 d and $C_0$ (Cd) = 0.1 mM. Similarly, the effect of $C_0$ (Sb) on Sb adsorption by fungi was investigated under the conditions of $pH_0$ = 4 and T = 7 d. The effect of adsorption time (T) on Sb adsorption by fungi was investigated under the conditions of $pH_0$ = 4 and $C_0$ (Sb) = 1 mM. The effect of the initial pH on Sb adsorption by fungi was studied under the conditions of T = 7 d and $C_0$ (Cd) = 0.1 mM. All experiments were repeated three times.

### 2.5. Statistical Analysis

All experiments were performed with three repetitions. SPSS 23.0 (SPSS, Chicago, IL, USA) was used to analysis the experimental data and one factor analysis of variance (One Way ANOVA) was employed. In all analyses, the standard deviation (S.D.) represented sample variability, and the significant differences was set to $p < 0.05$.

## 3. Results

### 3.1. The Phylogenetic Relationship and Morphological Characteristics of XK10

A Cd- and Sb-tolerant strain was screened out, named XK10. The surface of XK10 was non-uniform with a raised portion in the middle. Its colony was large and dense, with a yellowish color spreading outward. The edge of the colony was white (Figure 2a). Figure 2b shows the morphology of mycelial pellets of XK10. The phylogenetic tree constructed by BLAST comparison and MrBayes method in NCBI database showed that XK10 was most closely related to Penicillium spp. (Figure 2c) and XK10 was identified as *Penicillium* spp. The IAA and siderophore production were not detected in XK10 and phosphate solubilization was 15.72 ± 4.89 mg/L.

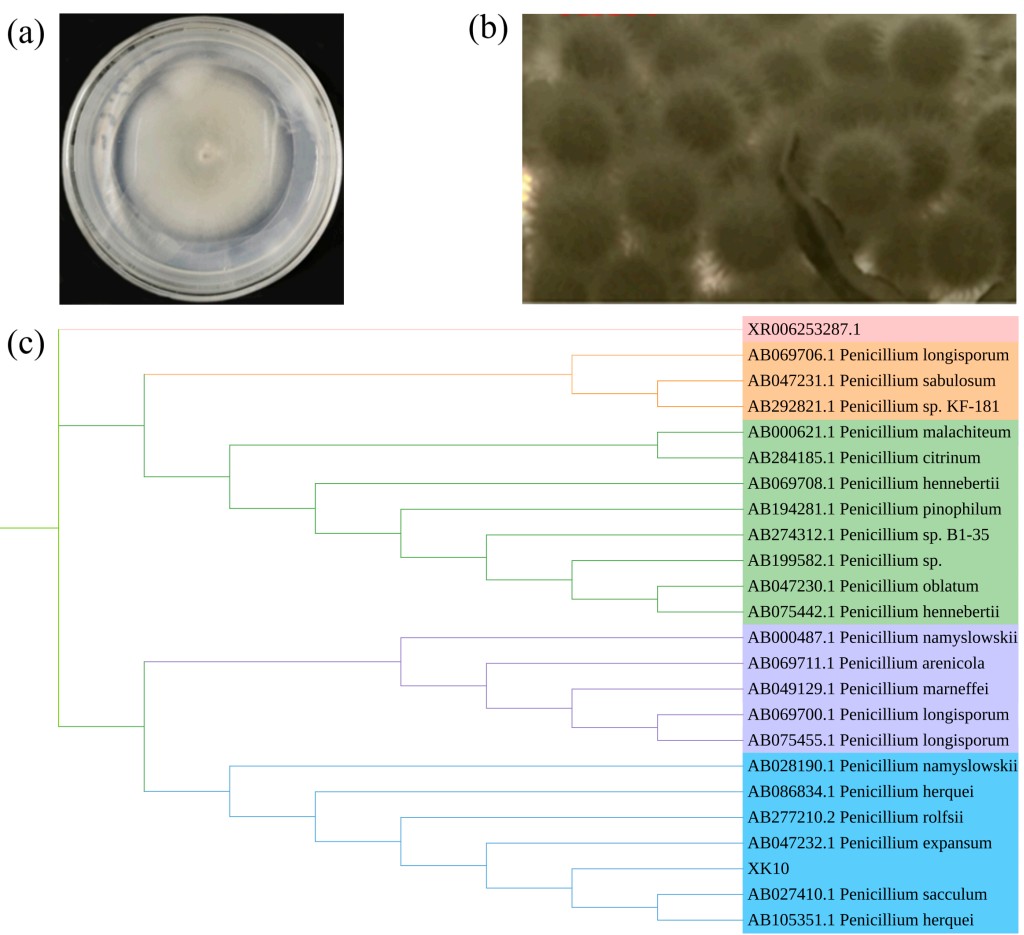

**Figure 2.** The morphology and phylogenetic tree of XK10. (**a**) Morphology of XK 10. (**b**) Morphology of mycelial pellets of XK10. (**c**) Phylogenetic tree of XK10.

### 3.2. Growth Curve of XK10

Microbial growth is generally divided into four phases: the lag phase; the logarithmic growth phase; the stabilization phase and the decay phase. The growth curve simulation showed that XK10 was fitted well with the theoretical model (Figure 3). XK10 was in the lag phase before 24 h, microbial biomass increased slowly before it adapted to the new environment. Then, the bacteria were active, and the growth rate was fast and entered the logarithmic growth phase. The dry weight increased significantly, and the growth rate of XK10 reached the maximum at 36 h. XK10 entered the stable phase after 96 h and the biomass was basically stable with a value of 0.25 g.

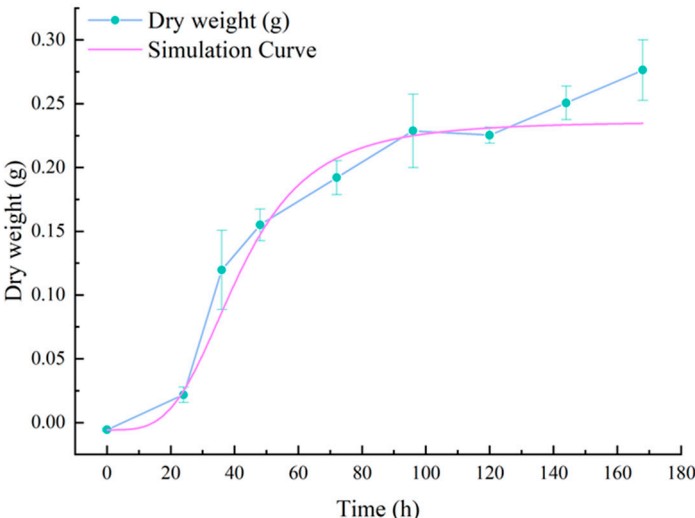

**Figure 3.** Growth curves of XK10.

### 3.3. Heavy Metal Tolerance Analysis of XK10

The biomass ($M_{LP}$) of XK10 at the logarithmic growth period was 0.1197 g (Figure 4). Cd tolerance studies of XK10 (Figure 4a) showed that after 1 d, compared with the control group, the biomass of XK10 was markedly reduced in the Cd-stressed group ($p < 0.05$), and the biomass of XK10 was the lowest at 2 mM Cd concentration, which was 0.0535. After 7 d, the biomass of XK10 was significantly lower than that of the control only at a Cd concentration of 2 mM ($p < 0.05$). However, the growth of XK10 was not restricted at the Cd concentrations of 0.1 mM and 0.5 mM. The inhibitory effect of heavy metal Sb on the growth of XK10 was less than that of Cd. After 7 d, the biomass of XK10 showed an increasing trend with the increase in Sb solution concentration (Figure 4b). When the Sb concentration reaches 20 mM, XK10 still grows normally, indicated that XK10 has a strong resistance to antimony.

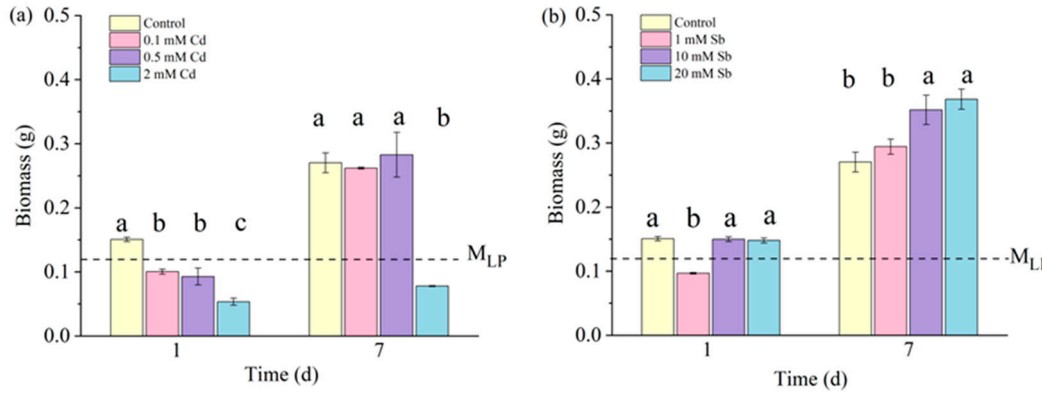

**Figure 4.** Growth of XK10 under different concentrations of Cd and Sb. (**a**) The biomass of XK10 in different Cd solution; (**b**) The biomass of XK10 in different Sb solution. $M_{LP}$ represented the biomass of XK10 in the logarithmic growth phase. Results are expressed as mean ± SD of three replicates. Different letters indicate significant differences between the same indicators ($p < 0.05$).

### 3.4. Effect of Initial Cd and Sb Concentrations on the Adsorption of XK10

The study of the adsorption of XK10 by the initial Cd concentration (Figure 5a) showed that the biomass of XK10 decreased gradually with the increase in the initial Cd concentration. When the Cd concentration was 0.5 mM, the biomass of XK10 was the lowest, which was 0.17 g. As the initial Cd concentration increased, the medium pH of XK10 was first stable ($0 < C_0 < 0.3$ mM) and then increased ($0.3 < C_0 < 0.5$ mM). At $C_0 = 0.5$ mM, the pH

of the medium increased to 4.4, indicating that high concentrations of Cd inhibited acid production by XK10 (Figure 6a). The removal rate of Cd by XK10 increased significantly with the increase in the initial Cd concentration ($p < 0.05$). The highest removal rate of Cd by XK10 was 14.7% when $C_0$ was 0.5 mM (Figure 7a).

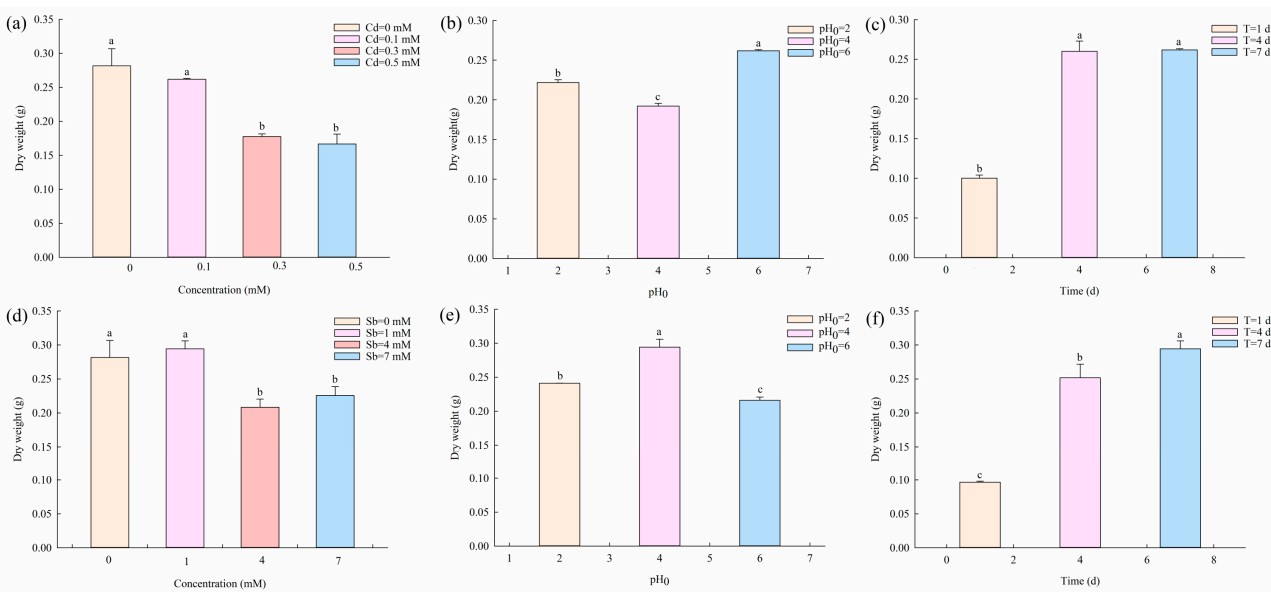

**Figure 5.** Effects of different factors on the dry weight of XK10 in the adsorption process. (**a**) Initial concentration for Cd adsorption; (**b**) Initial $pH_0$ for Cd adsorption; (**c**) Adsorption time for Cd adsorption; (**d**) Initial concentration for Sb adsorption; (**e**) Initial $pH_0$ for Sb adsorption; (**f**) Adsorption time for Sb adsorption. Results are expressed as mean ± SD of three replicates. Different letters indicate significant differences between the same indicators ($p < 0.05$).

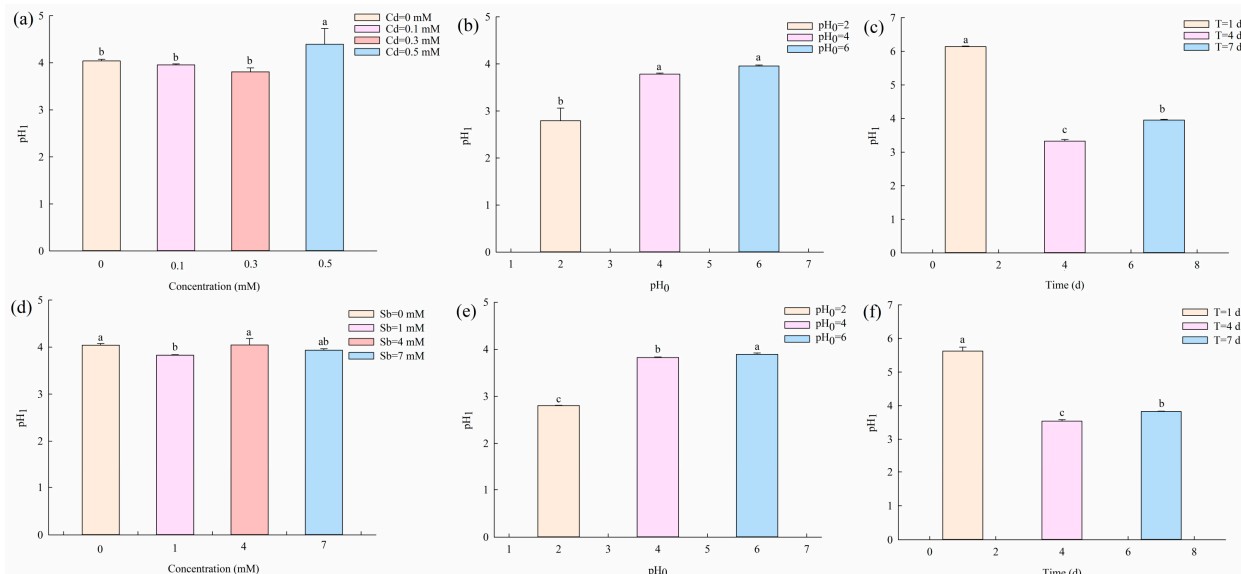

**Figure 6.** Effects of different factors on the $pH_1$ of XK10 in the adsorption process. (**a**) Initial concentration for Cd adsorption; (**b**) Initial pH0 for Cd adsorption; (**c**) Adsorption time for Cd adsorption; (**d**) Initial concentration for Sb adsorption; (**e**) Initial $pH_0$ for Sb adsorption; (**f**) Adsorption time for Sb adsorption. Results are expressed as mean ± SD of three replicates. Different letters indicate significant differences between the same indicators ($p < 0.05$).

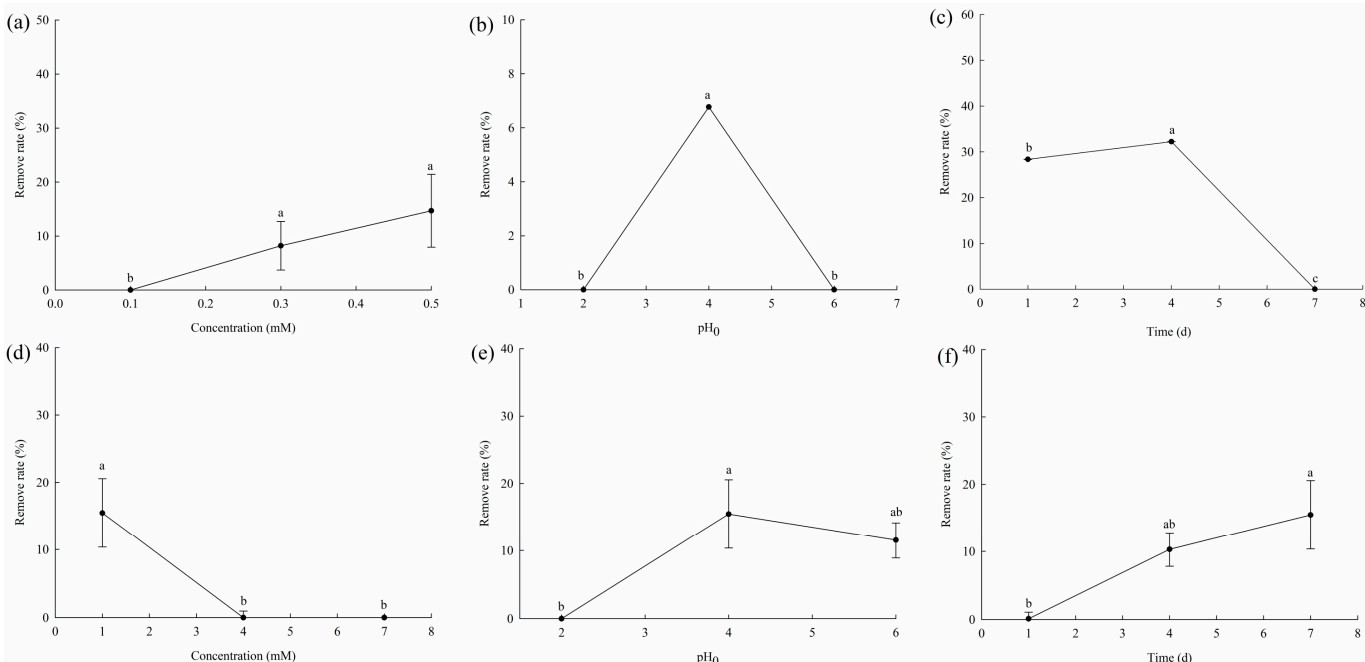

**Figure 7.** Effects of different factors on the remove rate of XK10 in the adsorption process. (**a**) Initial concentration for Cd adsorption; (**b**) Initial $pH_0$ for Cd adsorption; (**c**) Adsorption time for Cd adsorption; (**d**) Initial concentration for Sb adsorption; (**e**) Initial $pH_0$ for Sb adsorption; (**f**) Adsorption time for Sb adsorption. Results are expressed as mean ± SD of three replicates. Different letters indicate significant differences between the same indicators ($p < 0.05$).

The adsorption study of the initial Sb concentration on XK10 (Figure 5d) showed that with the increase in the initial Sb concentration, the biomass of XK10 gradually decreased, and the pH of the medium had little effect, the pH was stable at about 3.96 (Figure 6d). With the increase in Sb concentration, the removal rate of Sb by XK10 decreased obviously ($p < 0.05$). It dropped to 0% at an Sb concentration of 4 mM. When the Sb concentration was 1 mM, XK10 had the best Sb removal effect, which was 15.5% (Figure 7d).

### 3.5. Effect of Initial pH on the Adsorption of Cd and Sb by XK10

The effect of the initial pH on the adsorption of Cd by XK10 (Figure 5b) showed that the biomass of XK10 first decreased and then increased significantly with the increase in the initial pH ($p < 0.05$). At $pH_0 = 4$, the biomass of XK10 was the smallest at 0.19 g. At $pH_0 = 6$, the biomass of XK10 was the largest, reaching 0.26 g. The pH of the medium increased gradually with the increase in the initial pH. When $pH_0 = 6$, the pH was the largest, reaching 3.96 (Figure 6b). With the increase in the initial pH, the removal rate showed a trend of increasing first and then decreasing significantly ($p < 0.05$). When $pH_0 = 4$, the removal rate was the largest, which was 6.7% (Figure 7b).

The initial pH on the adsorption of Sb by XK10 (Figure 5e) showed that the biomass of XK10 increased first and then decreased significantly with the increase in the initial pH ($p < 0.05$). When $pH_0 = 4$, the biomass was the largest, reaching 0.29 g. At $pH_0 = 6$, the biomass was the smallest at 0.22. The pH of the medium showed a significant upward trend with the increase in the initial pH. When $pH_0 = 6$, the pH of the medium was the largest, reaching 3.89 (Figure 6e). When the initial pH of the solution was 2–4, the removal rate of antimony by XK10 increased significantly ($p < 0.05$), and when $pH_0 = 4$, the removal rate of antimony by XK10 was the highest, reaching 15.5%. When the initial $pH_0$ was 4–6, there was no significant difference in the removal rate of antimony by fungi (Figure 7e).

### 3.6. Effect of Adsorption Time on the Adsorption of Cd and Sb by XK10

The adsorption time study on the adsorption of Cd by XK10 (Figure 5c) showed that the biomass of XK10 increased significantly and then remained unchanged with the increase in adsorption time, and the biomass increased by 0.16 g. The pH of the medium showed a trend of decreasing first and then increasing with time ($p < 0.05$). After 4 days of adsorption, the pH dropped by 2.8, indicating that XK10 may have produced acidic species during the adsorption process. After 4–7 days of adsorption, the pH of the medium increased significantly ($p < 0.05$), reaching 4.0 (Figure 6c). The removal rate showed a trend of increasing first and then decreasing with time ($p < 0.05$). When $pH_0 = 4$, the removal rate was the largest, reaching 32.2%. When $pH_0 = 6$, it dropped to 0. It shows that after 4–7 days of adsorption, heavy metals may have toxic effects on XK10, causing the strain to die, resulting in a significant reduction in its removal rate of Cd (Figure 7c).

The adsorption time of XK10 on the adsorption of Sb (Figure 5f) showed that with the increase in adsorption time, the biomass of XK10 showed a significant increase trend ($p < 0.05$). At 7d, the biomass was the largest, which was 0.29 g. The pH of the medium showed a trend of firstly decreasing and then increasing significantly with the increase in time ($p < 0.05$). After 4 days of adsorption, the pH of the medium dropped to a minimum of 3.5. After 4–7 days of adsorption, the pH increased by 0.29 (Figure 6f). The removal rate increased with time and reached a maximum of 15.5% at 7 d (Figure 7f).

### 3.7. Adsorption Isothermal Model

The Langmuir model and Freundlich model are two extensively used mathematical models for simulating adsorption processes [34]. Both of these absorption models were employed in the study. The Langmuir isothermal model assumes that the adsorption sites are uniformly distributed on the adsorbent surface and that the contaminants form a monolayer on the surface. The Freundlich model assumes that monolayer adsorption has a homogeneous energy density distribution of active sites accompanied by interactions between the adsorbed molecules. We simulated single adsorption of Cd and Sb ions by fungi using both models, and the results are shown in Table 1. From Table 1, it can be seen that the $R^2$ of the Freundlich model for both Sb(III) and Cd(II) adsorption of the strain was larger than that of the Langmuir model. Therefore, we suggest that the biosorption of Sb(III) and Cd(II) by XK10 follows more closely the Freundlich model.

**Table 1.** Parameters of isotherm model on biosorption of Cd(II) and Sb(III) by XK10.

| The Name of the Strain | Cd(II) Single Adsorption (XK10) | Sb(III) Single Adsorption (XK10) |
|---|---|---|
| Langmuir model $K_L$ | 0.2153 | 0.0325 |
| $q_{max}$ | 0.28 | 4.81 |
| $R^2$ | 0.3307 | 0.5081 |
| Freundlich model $K_F$ | 0.0674 | 0.6593 |
| $n$ | 2.1329 | 2.7155 |
| $R^2$ | 0.4265 | 0.6387 |

## 4. Discussion

In this study, the tolerance XK10 to Cd in solid medium was greater than 5 mM and to Sb was greater than 7 mM. In liquid medium, XK10 could grow normally even when the concentration of Sb reached 20 mM, indicating that XK10 has a strong tolerance to Sb. Under normal conditions, heavy metals might poison the cells of soil microorganisms, slow their growth and cause microbial death. However, over time, some microorganisms may adapt to the heavy metals to survive and gradually become the dominant population in the soil [14,35]. In the presence of high concentrations of heavy metals, some microorganisms with a certain tolerance to heavy metals survive, and some reduce toxicity through biotransformation or metabolic activities. The tolerance of microorganisms to heavy metals can be determined by the minimum inhibitory concentration (MIC). MIC refers to the minimum

concentration of heavy metals in which a single microbial colony cannot grow. Its determination can be divided into liquid culture and solid culture, and the MIC values under different culture conditions are very different [36]. For example, *Penicillium chrysogenum* was significantly less tolerant of the heavy metal cadmium when grown on solid media than on liquid media [37]. In general, the higher the tolerance of the fungus, the greater its potential for application for environmental remediation. Currently, *Penicillium* has been shown to be tolerant to a variety of heavy metals [15]. The results of the analysis of the minimum inhibitory growth concentrations of different microorganisms for Cd and Sb showed that XK10 was more tolerant to both Cd and Sb than most microorganisms (Table 2). Therefore, it has a strong application potentiality.

**Table 2.** The minimum inhibitory concentrations of Cd and Sb by different microorganisms.

| Minimum Growth Inhibition Concentration | Adsorbent Material | MIC (Solid Culture) (mM) | MIC (Liquid Culture) (mM) | Reference Literature |
|---|---|---|---|---|
| Cd | XK10 | >5 | 2 | |
| | Uncultured *Westerdykella* | >1.3 | | [10] |
| | *Pseudomonas* sp. | 4 | | [30] |
| | *Pseudomonas azotoformans* | 0.9 | | [32] |
| | *Penicillium* spp. | | 1.3 | [38] |
| Sb | XK10 | >7 | >20 | |
| | *Cupriavidus* sp. | 6 | | [39] |
| | *Bacillus* sp. | 5.5 | | [39] |
| | *Penicillium* sp. | | 4.9 | [40] |

The initial Cd concentration in the solution affected the growth and adsorption efficiency of the strain. The removal of Cd by XK10 was not high at low concentrations and gradually increased with increasing concentrations, probably due to the higher initial Cd concentration providing the driving force to overcome the mass transfer resistance of the biosorption between the biosorption media [33]. The best removal of Sb by XK10 was achieved at an Sb concentration of 1 mM. The removal of Sb by XK10 decreased significantly with the increasing Sb concentration. It decreased to 0% at an Sb concentration of 4 mM. When the Sb concentration was too high, the repulsion between heavy metal ions in the solution increased, the removal sites on the cell membrane were saturated and some of the bacteria were lysed or autolyzed, thus leading to poor removal [41].

Solution pH can affect biosorption by influencing the activity of adsorption sites on the surface of the strain and the morphology of heavy metal ions [42,43]. At an initial pH of 2–4, the adsorption rates of XK10 for both Cd and Sb increased significantly and reached the maximum adsorption rate when the pH was 4. Poor protonation or ionization of functional groups at low pH leads to weak complexation stability of metal ions [44]. When the initial pH increases, the medium pH of XK10 increases (medium pH < 5), the negative charge on the surface of the biosorbent increases, the adsorption effect is enhanced, and positively charged $Cd^{2+}$ and $Sb(OH)^{2+}$ are adsorbed and the adsorption rate increases [45]. At an initial pH of 4–6, XK10 showed a decreasing trend in the adsorption rates of both Cd and Sb. At this time, Sb in solution exists in the form of $H_2SbO^{3-}$ or $Sb(OH)^{4-}$, and the negatively charged Sb and the negative charge on the surface of the biosorbent repel each other, and the adsorption rate of Sb decreases significantly.

The adsorption time is one of the important factors of the biosorption process [46]. For Cd, the biomass of XK10 showed a predominantly increasing trend at first and then leveling off as the adsorption time increased. Due to the massive bioactive adsorption sites on the fungal cell surface at the beginning of adsorption, the biosorption rate of metal ions is rapid, and then saturation was reached. The maximum Cd removal by XK10 occurred at the adsorption time of day 4. At the adsorption time of 4–7 days, the heavy metal may



have had a toxic effect on XK10, causing the strain to die, resulting in a significant decrease in the fungal removal of Cd. For Sb, the removal of Sb by XK10 continued to increase as the adsorption time increased, indicating that XK10 has a strong adsorption capacity for Sb.

The isothermal adsorption model can be developed to further understand the biosorption process [47]. In this study, the fits of Langmuir and Freundlich isotherms were used to simulate the biological processes of fungal adsorption of Cd and Sb. By comparing the linear correlation coefficients of the two models, it was found that the adsorption of Cd and Sb by XK10 is more consistent with the Freundlich model. The adsorption process is monolayer adsorption, the energy distribution on the active site is uneven and there is interaction between the adsorbed molecules.

## 5. Conclusions

The adsorption effect of microorganisms can be affected by several factors, such as the concentration of metal ions in the environment, time, temperature, pH, etc. However, there are relatively few studies and applications on the effect of microorganisms on the removal of heavy metal Sb. In this study, a Cd- and Sb-tolerant fungus named *Penicillium* spp. XK10, was isolated from the tailings slag. It has strong tolerance and certain adsorption capacity for Sb and Cd. At $pH_0 = 6$, $C_0$ (Cd) = 0.1 mM, and an adsorption time of 4 days, the maximum removal rate of cadmium by XK10 was 32.2%. Under the conditions of $pH_0 = 4$, T = 7 d, and the initial antimony concentration of 1 mM, the removal rate of antimony by XK10 was the highest, which was 15.5%. The initial pH of heavy metal solution, adsorption time, initial Cd concentration and initial Sb concentration had significant effects on the growth of XK10, pH of the medium and the removal rate of Sb and Cd. Acid was produced by XK10 during the adsorption of Sb and Cd. The weak acidic environment prevailing in the mine will promote the adsorption of Sb and Cd by XK10. Therefore, it provides a potential material for bioremediation of Sb and Cd pollution in mines.

**Author Contributions:** Conceptualization, Y.H. and C.L.; methodology, C.L., Y.H. and Z.X.; software, Y.H.; validation, W.Z. (Wan Zhang) and J.H.; formal analysis, Y.H.; investigation, C.L.; resources, Y.Z. and Z.X.; data curation, Z.X.; writing—original draft preparation, Y.H.; writing—review and editing, W.Z. (Wan Zhang) and Z.X.; visualization, Y.H., C.L. and Z.X.; supervision, Z.S., W.Z. (Weiping Zhao); project administration, Y.Z. and Z.X.; funding acquisition, Y.Z. and Z.X. All authors have read and agreed to the published version of the manuscript.

**Funding:** This research was funded by the Key Projects of National Forestry and Grassland Bureau (201801), China Postdoctoral Science Foundation (2020M683592), and Open Fund of Key Laboratory of Microbial Resources Collection and Preservation, Ministry of Agriculture and Rural Affairs (KLMRCP2021-07). The funding bodies played no role in the design of the study and collection, analysis, and interpretation of data and in writing the manuscript.

**Institutional Review Board Statement:** Not applicable.

**Informed Consent Statement:** Not applicable.

**Data Availability Statement:** Data are publicly available with accession number OP692700 and interested parties may contact the authors for more information.

**Conflicts of Interest:** The authors declare no conflict of interest.

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
