# Peer review of "Penicillium spp. XK10, Fungi with Potential to Repair Cadmium and Antimony Pollution"

_applsci, doi:10.3390/app13031228_

Round 1

Reviewer 1 Report

Dear Jean Deng

Assistant Editor

Dear Author,  

I read the manuscript applsci-2076437-peer-review-v1: ”  Penicillium spp. XK10, fungi with potential to repair cadmium and antimony pollution”, and I present below a few of my observations:

1.     A clearer delimitation of the purpose of this study is necessary, at the end of the Introduction.

2.     Subchapter 2.5. Langmuir and Freundlich do not represent statistical analysis!!!!! Based on these adsorption isotherms, the process is modeled to calculate characteristic quantitative quantities and to appreciate the adsorption mechanism.

3.     How can this difference in the behavior of the microbial biomass without Cd and compared to Sb be explained? For Sb, a clearly superior behavior appears!

The work deals with an interesting topic, but there are a few small changes to be made in order to be published.

Sincerely yours,

Author Response

Response to Reviewer 1 Comments

Thank you very much for the professional opinions of the reviewers. Your suggestion are very helpful to improve my manuscript. All changes in the article have been marked and the responses were addressed as follows.

Point 1:  A clearer delimitation of the purpose of this study is necessary, at the end of the Introduction.

Response 1: Thank you for your comments. We have added the purpose of this study at the end of the Introduction in Line 81-94.

Point 2: Subchapter 2.5. Langmuir and Freundlich do not represent statistical analysis!!!!! Based on these adsorption isotherms, the process is modeled to calculate characteristic quantitative quantities and to appreciate the adsorption mechanism.

Response 2: Thank you for your comments. We have corrected the 2.5 and 3.7 section to clarify the interpretation of the adsorption mechanismand.

Point 3: How can this difference in the behavior of the microbial biomass without Cd and compared to Sb be explained? For Sb, a clearly superior behavior appears!

Response 3: Thank you for your comments. We have explained this behavior in Line 372-384.

We would like to thank you again for taking the time to review our manuscript.

Reviewer 2 Report

A very meaningful work. The authors prepared well and gave a very detail introduction about the work. There are several questions for the authors to explain in the Method Part:

#Comment 1:

Line 75-76: It will be better if you add some information on the history of the antimony mine. 

#Comment 2:

Line 76-78: what is the value of "the soil contamination risk control standards"? Please add and explain.

#Comment 3:

Line 79-81: Please add the detail process of soil sample collection. For example, when did you conduct the soil sampling? how many sites? how many replicates? et al.

Author Response

Response to Reviewer 2 Comments

Thank you very much for the professional opinions of the reviewers. Your suggestion are very helpful to improve my manuscript. All changes in the article have been marked and the responses were addressed as follows.

Point 1:  Line 75-76: It will be better if you add some information on the history of the antimony mine.  

Response 1: Thank you for your comments. We have added some information on the history of the antimony mine in Line 97-99.

Point 2: Line 76-78: what is the value of "the soil contamination risk control standards"? Please add and explain.

Response 2: Thank you for your suggestion. The "soil contamination risk control standards" refers to the threshold value of heavy metals in soil specified locally. Considering that the above standards are only local significance, we delete the above statement in the revised manuscript.

Point 3: Line 79-81: Please add the detail process of soil sample collection. For example, when did you conduct the soil sampling? how many sites? how many replicates? et al.

Response 3: Thank you for your comments. More details of collected soil samples were added in Line 118-122.

We would like to thank you again for taking the time to review our manuscript.
